# Origin of acetylcholine antagonism in ELIC, a bacterial pentameric ligand-gated ion channel

Mykhaylo Slobodyanyuk [1,2,4], Jesús A. Banda-Vázquez [1,2,4], Mackenzie J. Thompson [3], Rebecca A. Dean [1,2], John E. Baenziger [3], Roberto A. Chica [1,2✉] & Corrie J. B. daCosta [1,2✉]

ELIC is a prokaryotic homopentameric ligand-gated ion channel that is homologous to vertebrate nicotinic acetylcholine receptors. Acetylcholine binds to ELIC but fails to activate it, despite bringing about conformational changes indicative of activation. Instead, acetylcholine competitively inhibits agonist-activated ELIC currents. What makes acetylcholine an agonist in an acetylcholine receptor context, and an antagonist in an ELIC context, is not known. Here we use available structures and statistical coupling analysis to identify residues in the ELIC agonist-binding site that contribute to agonism. Substitution of these ELIC residues for their acetylcholine receptor counterparts does not convert acetylcholine into an ELIC agonist, but in some cases reduces the sensitivity of ELIC to acetylcholine antagonism. Acetylcholine antagonism can be abolished by combining two substitutions that together appear to knock out acetylcholine binding. Thus, making the ELIC agonist-binding site more acetylcholine receptor-like, paradoxically reduces the apparent affinity for acetylcholine, demonstrating that residues important for agonist binding in one context can be deleterious in another. These findings reinforce the notion that although agonism originates from local interactions within the agonist-binding site, it is a global property with cryptic contributions from distant residues. Finally, our results highlight an underappreciated mechanism of antagonism, where *agonists* with appreciable affinity, but negligible efficacy, present as competitive antagonists.

[1] Department of Chemistry and Biomolecular Sciences, University of Ottawa, Ottawa, ON, Canada. [2] Centre for Chemical and Synthetic Biology, University of Ottawa, Ottawa, ON, Canada. [3] Department of Biochemistry, Microbiology, and Immunology, University of Ottawa, Ottawa, ON, Canada. [4] These authors contributed equally: Mykhaylo Slobodyanyuk, Jesús A. Banda-Vázquez. ✉email: rchica@uottawa.ca; cdacosta@uottawa.ca

Agonists bind to receptors and activate them, whereas antagonists bind to receptors and inhibit them. These fundamental concepts of pharmacology have been studied for more than a century[1]. For ligand-gated, or agonist-activated, ion channels, activation involves opening of an intrinsic ion-conducting pore[2]. Agonists open the channel because they have a higher affinity for the open/activated state than the closed/resting state, and thus their binding shifts the equilibrium towards the open/activated state[3]. Antagonists, on the other hand, inhibit channel opening because they have a higher affinity for the closed/resting state. Distilled in this way, drug action boils down to interactions of the agonist, or antagonist, with specific conformations of their receptor.

ELIC (Fig. 1a, b) is a cation-selective homopentameric ligand-gated ion channel from the gram-negative bacterium, *Erwinia chrysanthemi* (subsequently renamed to *Dickeya dadantii*), which is part of a larger superfamily of pentameric ligand-gated ion channels that were first identified in vertebrates[4–6]. The family includes cation-selective channels activated by acetylcholine (ACh) or serotonin (5-hydroxytryptamine), as well as anion-selective channels activated by γ-aminobutyric acid (GABA) or glycine[7]. Known agonists of ELIC include cysteamine and propylamine, both of which are primary amines[8]. Testifying to its homology with eukaryotic pentameric ligand-gated ion channels, ELIC also binds the vertebrate neurotransmitters, GABA[8,9] and ACh[10]. For ELIC, GABA acts as an agonist[8], whereas ACh behaves as a competitive antagonist[10]. Structures of ELIC in complex with either GABA[9] (Fig. 1c) or ACh[10] (Fig. 1d) show that both ligands bind similarly to the same site, however, the binding of ACh stabilizes ELIC in a conformation described as being on the verge of activation, where the channel pore remains too constricted to conduct ions[10]. Remarkably, the ACh derivative, 2-dimethylaminoethylacetate (DMAEA), which differs from ACh by a single methyl group, is able to activate ELIC[10]. These observations suggest that subtle structural alterations of the ELIC agonist-binding site might be sufficient to tip the balance and allow ACh to activate an engineered ELIC.

Given that ELIC binds ACh and is structurally homologous to acetylcholine receptors (AChRs), we asked whether it is possible to convert ACh from an ELIC antagonist into an ELIC agonist by substituting AChR residues into ELIC. Using available crystal structures[9–11], and statistical coupling analysis[12], we identify candidate residues that could be involved in agonism, and which differ between ELIC and AChRs. Substitution of individual AChR residues into ELIC did not convert ACh into an ELIC agonist, but allowed us to identify two residues that reduced the sensitivity of ELIC to ACh inhibition. When these two substitutions are combined, the double mutant is no longer inhibited by ACh. Thus, making the ELIC agonist-binding sites more AChR-like does not install ACh agonism, but instead knocks out ACh antagonism, with the simplest interpretation being that these substitutions reduce ACh affinity. Our results reinforce the notion that the mechanisms by which agonists and antagonists elicit their responses is a global property of the protein, dependent upon contributions from residues distant from the agonist-binding site. The cryptic nature of these long-range contributions complicates attempts to swap agonist specificities between homologous receptors. Furthermore, our findings speak to an underappreciated mechanism of antagonism, where an *agonist* with appreciable affinity, but negligible efficacy, such as ACh for ELIC, presents as a competitive antagonist.

## Results

### Activation and inhibition of ELIC.
To confirm that ACh inhibits agonist-activated currents through ELIC, we used whole-cell two-electrode voltage clamp, an electrophysiological method that has been used extensively to study both agonism and antagonism in a variety of pentameric ligand-gated ion channels, including ELIC[10]. We began by assessing the potency of two well-known ELIC agonists, cysteamine and GABA. Consistent with previous findings[8], wild-type ELIC heterologously expressed in *Xenopus laevis* oocytes produced a dose response with half maximal effective concentrations ($EC_{50}$) of 0.6 mM and 2.7 mM for cysteamine and GABA, respectively (Fig. 1e–g; Table 1). To assess the ability of ACh to inhibit cysteamine or GABA-activated currents, we then added increasing concentrations of ACh while maintaining the cysteamine or GABA concentration at their respective $EC_{50}$ (Fig. 1h–j). In each case, this led to an ACh-dependent decrease in the agonist-activated peak current. The measured half maximal inhibitory concentrations ($IC_{50}$) were not significantly different for ACh inhibition of cysteamine and GABA-activated currents (Fig. 1j; Table 1), and thus ACh was equally potent an inhibitor in the presence of either agonist. Since GABA-activated currents exhibited slower macroscopic activation/deactivation kinetics (Supplementary Fig. 1), and also required higher GABA concentrations for activation owing to its lower potency, we proceeded with cysteamine as the agonist for subsequent experiments.

ELIC has been crystallized in complex with GABA (PDB ID: 2YOE)[9], and separately in complex with ACh (PDB ID: 3RQW)[10]. Both ligands share a similar binding pose within the agonist-binding sites, and many of the residues that stabilize each ligand adopt a similar side chain conformation in the two crystal structures (Fig. 1). The question then is: why does ACh fail to activate ELIC? Furthermore, can ACh agonism be installed into ELIC by substituting AChR residues into the agonist-binding site, thereby providing insight into the determinants of ACh agonism?

### Identifying candidate residues involved in agonism.
ELIC in complex with ACh has been characterized as being on the verge of activation[10]. Consistent with this observation, 2-dimethylaminoethylacetate (DMAEA), a tertiary amine derivative of ACh that lacks a single methyl group, can activate ELIC[10]. These results suggests that subtle alterations of the ELIC agonist-binding site via mutagenesis might be enough to tip the balance towards ACh agonism. To identify candidate residues for mutation, we took a three-pronged approach. First, we compared the structures of ELIC in complex with either ACh or GABA and identified residues in direct contact with the ligands, or whose side chains adopted alternate conformers in the two crystal structures (e.g. L178). Second, we identified residues in the ELIC agonist-binding site whose identity differed from the corresponding residues in structurally related AChRs (e.g. F133), for which ACh is a full agonist. Lastly, hypothesizing that evolutionary information embedded in related proteins might be able to point us towards additional residues important for agonism, we performed statistical coupling analysis[13]. Statistical coupling analysis identifies groups of residues whose conservation and substitution patterns are statistically correlated within a multiple sequence alignment. These groups of residues, often called "protein sectors", have been implicated in protein functions specific to the family of homologous proteins being studied[12].

Statistical coupling analysis requires a comprehensive multiple sequence alignment that contains a diverse set of homologous sequences. Rather than using the typical sequence-based approach for retrieving homologous sequences (i.e., BLAST search)[14], we employed a structure-guided strategy[15] using, as templates, (1) the 2.60 Å crystal structure of an acetylcholine binding protein (AChBP) in complex with ACh (PDB ID: 3WIP)[11], and (2) the 2.70 Å cryo-electron microscopy structure

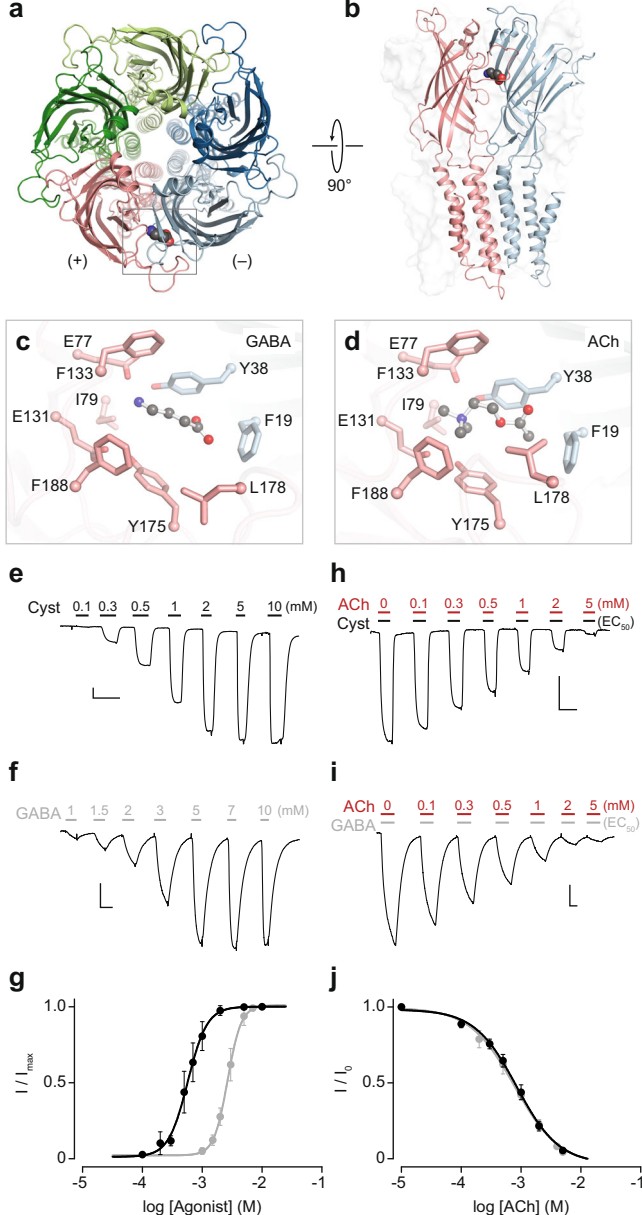

**Fig. 1 ACh binds to the agonist-binding site of ELIC and competitively inhibits agonist activated currents. a, b** Crystal structure (PDB ID: 2YOE)[9] of ELIC in complex with the agonist GABA (spheres), where each of the five identical subunits are shown in a cartoon representation. Box indicates a single agonist-binding site at the interface between two subunits. Close-up views of the agonist binding site in complex with **c** GABA (PDB ID: 2YOE)[9] and **d** ACh (PDB ID: 3RQW)[10], with residues on the principal face (+; pink) and complementary face (−; cyan) highlighted as sticks. Ligands are shown as ball and sticks. Representative whole-cell traces for **e** cysteamine-activated (Cyst) and **f** GABA-activated ELIC, along with **g** the associated dose response curves for cysteamine (black circles) and GABA (gray circles). Representative whole-cell traces for ACh-inhibited ELIC activated by an $EC_{50}$ concentration of **h** cysteamine (0.6 mM) or **i** GABA (2.7 mM), as well as **j** the dose response curve for ACh inhibition in the presence of either cysteamine (black circles) or GABA (gray circles) at their respective $EC_{50}$. Agonist (**e**, **f**) and antagonist concentrations (**h**, **i**), both in mM, are indicated above each peak with the duration of application shown as black, gray, and red bars for cysteamine, GABA, and ACh, respectively. In (**e**, **f**, **h**, **i**) the x- and y-axis of each scale bar corresponds to 1 min and 0.5 μA, respectively. Error bars in the dose response curves represent one standard deviation from the mean, obtained from a minimum of 5 independent oocyte replicates.

of the human α7 AChR (α7) in complex with epibatidine and PNU-120596 (PDB ID: 7KOX)[16](Methods). The benefit of this structure-guided strategy is that the retrieved sequences not only share homology with the templates, but are also likely to adopt their backbone conformations, which in this case is that of AChBP bound to ACh, or α7 bound to epibatidine. If the conformation of AChBP bound to ACh, or α7 bound to epibatidine, resemble that of an ACh-activated AChR, then the retrieved sequences that form the basis of each statistical coupling analysis are likely to contain information relevant to ACh agonism. The same approach was successful in engineering ligand specificity in a periplasmic binding protein[17].

Our first statistical coupling analysis was based upon the AChBP from *Lymnaea stagnalis*, which is a soluble homo-pentameric protein that is homologous to the extracellular domain of both ELIC (Supplementary Figs. 2, 3) and related AChRs[18,19]. AChBPs have been used extensively as structural surrogates for understanding the atomic basis of ligand recognition, as well as the associated structural rearrangements that occur in the AChR extracellular domain upon ligand binding[11,20–24]. This consideration, along with the availability of a 2.60 Å resolution X-ray crystal structure[11], made the *Lymnaea stagnalis* AChBP in complex with ACh an attractive candidate for our structure-guided approach. Nevertheless, AChBPs lack both transmembrane and cytoplasmic domains, and although a chimeric construct in which the AChBP from *Lymnaea stagnalis* was coupled to a 5-HT$_{3A}$ receptor pore could be activated by ACh, it had to be extensively engineered[25], indicating that elements necessary for translating ACh-binding into channel activation might not be fully preserved in AChBPs. To overcome this potential limitation, we performed a second statistical coupling analysis based upon the 2.70 Å resolution cryo-electron microscopy structure of the human α7 AChR in complex with both epibatidine and the positive allosteric modulator, PNU-120596[16]. Like ACh, epibatidine is a full α7 agonist, and this ternary α7/epibatidine/PNU-120596 complex is thought to represent an activated/open conformation of the α7 AChR[16]. Combined, these two statistical coupling analyses should provide insight into residues involved in ACh/agonist recognition and agonism in this family of homologous proteins.

Our AChBP and α7-based statistical coupling analyses each identified 3 protein sectors, which were mapped onto the structures and aligned sequences of ELIC, AChBP, and the human α7 AChR (Supplementary Fig. 2). The AChBP-based analysis revealed that, many of the residues within sector 2 (red) cluster together, in and around the agonist-binding site, suggesting that this sector may represent a set of residues that collectively participate in agonist recognition, and potentially downstream activation. Of the complete sets of residues in sector 2 (Table 2), only four (Y38, I79, E131, F133) overlap with residues that are in direct contact with ligand in the various ELIC crystal structures (Fig. 1c, d and Fig. 2a). Gratifyingly, despite one of the residues (Y38) being in a separate sector, the same four residues were identified in our α7-based analysis. Of these four residues, we chose not to substitute E131 to preserve a strong interaction with bound ACh, as originally noted by Pan et al.[10]. Furthermore, we chose to substitute A75, a second-shell residue identified in both the AChBP and α7-based statistical coupling analyses. Even though A75 does not directly interact with the either GABA or ACh, this residue is a known determinant of agonist binding in the human muscle-type AChR[26]. Given that ACh-bound ELIC is thought to be on the verge of activation, we substituted residues found in the human α7 AChR (also conserved in the *Lymnaea stagnalis* AChBP) at these corresponding four sites into ELIC (Supplementary Fig. 3),

**Table 1 Activation and inhibition parameters for wild-type ELIC and its variants.**

| Activation | | | | Inhibition | | |
|---|---|---|---|---|---|---|
| ELIC | EC$_{50}$ (mM) | Hill coefficient | $n$ | ACh IC$_{50}$ (mM) | Hill coefficient | $n$ |
| WT (GABA) | 2.7 ± 0.3[b] | 4.0 ± 0.7 | 6 | 1.1 ± 0.2[d] | −1.0 ± 0.3 | 5 |
| WT (Cysteamine) | 0.6 ± 0.1 | 2.6 ± 0.4 | 11 | 1.0 ± 0.2[d] | −1.1 ± 0.1 | 8 |
| Y38W | 0.5 ± 0.1 | 1.4 ± 0.2 | 8 | 0.4 ± 0.1[c] | −1.4 ± 0.1 | 8 |
| A75D | 1.6 ± 0.4[b] | 2.3 ± 0.5 | 9 | 7 ± 2[c] | −0.7 ± 0.1 | 6 |
| I79Y[a] | 14 ± 1[b] | 2.2 ± 0.3 | 6 | -- | -- | -- |
| F133W | 4.3 ± 0.6[b] | 2.3 ± 0.3 | 8 | 8 ± 2[c] | −1.1 ± 0.1 | 8 |
| A75D/F133W | 6.9 ± 0.7[b] | 3.8 ± 0.3 | 7 | -- | -- | -- |
| L240A (L9′A) | 0.15 ± 0.01[b] | 2.8 ± 0.1 | 8 | -- | -- | -- |
| L240A (L9′A) + A75D/F133W | 4.3 ± 0.6[b] | 3.7 ± 0.3 | 6 | -- | -- | -- |

All EC$_{50}$, IC$_{50}$ and Hill coefficient values are indicated as mean ± s.d., with the number of independent replicates indicated as $n$.
WT wild-type.
[a]Estimate of EC$_{50}$ and Hill coefficient are provided based on cysteamine application for 30 s in the range of 1.5 mM to 100 mM.
[b]$p < 0.001$ relative to WT-ELIC (Cysteamine) log(EC$_{50}$) calculated using a one-way ANOVA followed by Dunnett's post-hoc test.
[c]$p < 0.001$ relative to WT-ELIC (Cysteamine) log(IC$_{50}$) calculated using a one-way ANOVA followed by Dunnett's post-hoc test.
[d]log(IC$_{50}$s) not significantly different ($p > 0.05$) calculated using a one-way ANOVA followed by Dunnett's post-hoc test.

and then determined if any of the substitutions were enough to tip the balance in favour of ACh agonism.

All mutant ELIC receptors (Y38W, A75D, I79Y, and F133W) were expressed in oocytes, and their activity was analyzed using two-electrode voltage clamp. Mirroring the results with wild-type ELIC, increasing concentrations of cysteamine led to progressively larger peak currents for all single mutants (Supplementary Fig. 4). The EC$_{50}$ for cysteamine of the Y38W mutant was unchanged compared to that of the wild-type, while the A75D and F133W mutants exhibited 3-fold and 7-fold increases in their respective EC$_{50}$s for cysteamine (Fig. 2b; Table 1). The reduced sensitivity to cysteamine of the A75D mutant indicates a role for this residue in ELIC activation, despite it having no direct interaction with the agonist. Although the I79Y mutant expressed and was functional in oocytes, it exhibited slow macroscopic activation (Supplementary Fig. 5b), and thus a pseudo-dose response curve was obtained by applying each concentration of cysteamine for a fixed interval of 30 s (Fig. 2b; Supplementary Fig. 4b). Relative to wild-type, the I79Y substitution resulted in a 24-fold increase in the EC$_{50}$ for cysteamine (Table 1).

We next tested whether ACh could activate these mutant ELIC receptors. Successive addition of ACh at concentrations of 0.5 mM to 50 mM was, in each case, followed by application of 5 mM cysteamine (Supplementary Fig. 5). None of the single mutants, nor wild-type ELIC, produced any peak current when ACh was applied. However, the application of 5 mM cysteamine produced a robust peak current in each case (Supplementary Fig. 5). Given the absence of ACh agonism in these single mutants, we investigated whether their sensitivity to ACh inhibition was altered. Once again keeping the cysteamine concentration fixed at its EC$_{50}$, ACh produced a concentration-dependent inhibition of the agonist peak current for the three single mutants tested (Supplementary Fig. 6b, c, d). Relative to wild-type, there was an increase in the ACh IC$_{50}$ for the A75D mutant (7-fold), as well as the F133W mutant (8-fold; Fig. 2c; Table 1), indicating a decrease in the sensitivity of these mutants to ACh inhibition. By contrast, the ACh IC$_{50}$ for the Y38W mutant decreased 3-fold, indicating that this mutant was more sensitive to ACh inhibition. Note that the ACh IC$_{50}$ for the I79Y mutant was not obtained due to its slow macroscopic activation and apparent toxicity to the *Xenopus laevis* oocytes upon expression, which can be seen by a steadily decreasing baseline signal throughout the whole-cell experiment (Supplementary Figs. 4b, 5b).

**A double mutant ELIC is potentiated by acetylcholine.** Both the A75D and F133W individual substitutions led to a reduction in sensitivity to ACh inhibition (Fig. 2c). To determine if ACh inhibition could be further reduced, we combined the A75D and F133W substitutions to generate a double mutant. This double mutant was expressed in oocytes and produced a concentration-dependent response to cysteamine, with an EC$_{50}$ of 6.9 mM (Fig. 3a, b; Table 1). Interestingly, ACh failed to inhibit the cysteamine response in the A75D/F133W double mutant, in contrast to what was observed for wild-type ELIC and its single mutants. At high concentrations (50–100 mM), ACh appeared to potentiate, rather than attenuate, the cysteamine peak current (Fig. 3c, d).

The ability of ACh to potentiate cysteamine currents in the A75D/F133W double mutant was next explored using a range of cysteamine concentrations, corresponding to the EC$_{10}$, EC$_{20}$, EC$_{50}$ and EC$_{90}$, in the absence and then presence of 25 mM ACh. Directly comparing the cysteamine-activated peak current in the absence and then presence of 25 mM ACh, confirmed that in the double mutant ACh potentiated the cysteamine response (Fig. 3e, f). Furthermore, this potentiation was dependent upon the concentration of cysteamine, with currents elicited by lower concentrations of cysteamine being less potentiated by ACh. The amount of ACh potentiation ranged from approximately 15% to 50% of the total peak current, for cysteamine concentrations between EC$_{10}$ and EC$_{90}$, respectively. For comparison, we repeated this experiment with wild-type ELIC and each of the single mutants, and observed that for both wild-type and the Y38W mutant, cysteamine-activated currents were nearly fully inhibited by 25 mM of ACh (Fig. 3f; Supplementary Fig. 7). In the F133W and A75D single mutants, increasing the concentration of cysteamine reduced the degree of inhibition by 25 mM ACh. By contrast, for the A75D/F133W double mutant, increasing the concentration of cysteamine further amplifies the extent of potentiation by ACh.

Given that the extent of the potentiation by ACh was modest, and that the concentration of ACh used to elicit it was high (25 mM), we assessed whether the potentiation could have originated from indirect effects, such as altered osmolarity or a change in ionic strength affecting the equilibrium potential of permeant ions. To do so, we performed a series of co-application experiments where the osmolarity and/or the ionic strength surrounding the cysteamine-activated double mutant was increased (Supplementary Fig. 8). In the middle of a cysteamine pulse (at EC$_{50}$), either 50 mM ACh, 50 mM sucrose, or an

**Table 2 Sector residues identified in the AChBP and α7-based statistical coupling analyses.**

| Sector | Residues[a] |
|---|---|
| Blue (AChBP) | P20,T21, R23, V29, N37, E40, V41, N42, E43, I44, T45, N46, V50, V51, W53 |
| Red (AChBP) | W53, D85, Y89, S126, E131, S132, A134, C136, K139, I140, G141, S142, W143, T144, H145, S147, S151 |
| Green (AChBP) | V48, S67, P77, L86, A87, A88, Y113, P115, S116, I117, Q119, F121, T144 |
| Blue (α7 AChR) | Y14, R19, P20, V21, N23, S25, V30, L36, Q38, I39, D41, V42, D43, E44, K45, N46, Q47, T50, T51, N52, W54, W59, D61, W66, V68, G73, P80, W85, P87, D88, I89 |
| Red (α7 AChR) | Q38, D41, Y71, D88, Y92, A95, T105, D130, K144, G146, W148, G152, D156, R182, Y187, C189, C190, E192, P193, Y194, P195, D196, T198, T207, F229, E237, L241 |
| Green (α7 AChR) | Y14, D61, L64, Y71, R78, L90, L91, N93, S94, D96, Y117, P119, G121, I122, I129, W133, S149, Y150, L208, Y209, L215, C218, S222, L231, P232, G236, K238, I243, T244, F252, T263, S264, S276, M278, L292, Q293, H295 |

[a]AChBP numbering is according to PDB ID: 3WIP (chain A), and for the α7 AChR as in PDB ID: 7KOX (chain A).

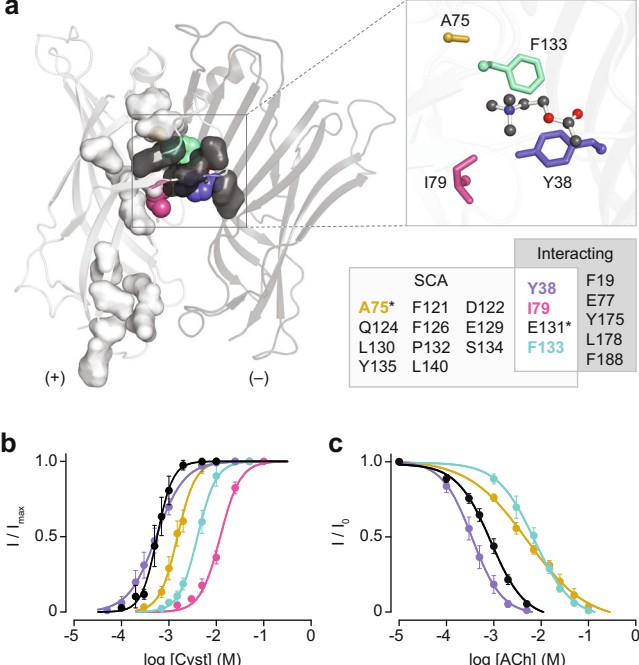

**Fig. 2 Identification of amino acids involved in ACh binding and antagonism. a** Crystal structure (PDB ID: 3RQW)[10] of ELIC in complex with ACh (ball and sticks), where the extracellular domain of the principal (+) and complementary (−) subunits are shown as cartoons and coloured light and dark grey, respectively. Residues identified by statistical coupling analysis or directly interacting in the structure are shown as light and dark grey surfaces, respectively. The four candidate residues for installing ACh agonism are shown as spheres, with A75, I79, and F133 on the principal subunit, and Y38 on the complementary subunit, coloured orange, magenta, green, and purple, respectively. These four residues are highlighted in the boxed inset on the right, and represented as sticks. A Venn diagram (below inset) shows the residues identified by statistical coupling analysis (SCA) and directly interacting with agonist in the structure (Interacting). E131 (asterisk) was not mutated since it has been identified as important for ACh binding in ELIC[10], while A75 (asterisk) was mutated since this position has been shown to be important for ACh binding in the AChR[26]. **b** Dose response curves of cysteamine-activated wild-type ELIC (black), as well as the Y38W (purple), A75D (orange), I79Y (magenta) and F133W (green) mutants. **c** Inhibitory ACh dose response curves in the presence of $EC_{50}$ cysteamine for the wild-type and mutants (excluding I79Y) are coloured as in (**b**). Error bars in (**b**, **c**) represent one standard deviation from the mean, obtained from a minimum of 6 independent oocyte replicates.

additional 50 mM sodium chloride (NaCl), was co-applied. Unlike what we observe with ACh, the presence of 50 mM sucrose did not potentiate the cysteamine response, demonstrating that the potentiation does not result from an increase in osmolarity alone (Supplementary Fig. 8a). Application of an additional 50 mM of extracellular NaCl however produced a small, but reproducible, degree of potentiation, likely due to the increased equilibrium potential for sodium ions (Supplementary Fig. 8b). However, by itself the potentiation by 50 mM NaCl was not enough to account for potentiation observed in the presence of 50 mM ACh. To assess whether a combined increase in osmolarity and ionic strength could account for the potentiation by ACh, we co-applied a solution containing both 50 mM sucrose as well as an additional 50 mM NaCl. The application of additional NaCl together with sucrose led to a degree of potentiation that was greater than that of NaCl alone, and resembled that observed with ACh, suggesting that the presence of sucrose compounded the potentiating effect of the NaCl (Supplementary Fig. 8c). Based on this observation, the simplest interpretation is that the potentiation observed in the presence of ACh is the result of an indirect effect, caused by the increased osmolarity and ionic strength due to the presence of high concentrations of ACh, an organic cation.

**Sensitivity of double mutant ELIC to structurally related amines.** Having established that ACh antagonism is abolished in the A75D/F133W double mutant ELIC, we next explored how the double mutant responded to structurally related amines, several of which are known ELIC agonists. In general, ELIC is activated by small, unbranched, primary amines, such as cysteamine, propylamine, and GABA[8]. While the double mutant is still activated by cysteamine, GABA fails to elicit a response, even at concentrations as high as 25 mM (Fig. 4c). The tertiary amines, DMAEA and the related trimethylamine, both activate the double mutant and wild-type ELIC (Fig. 4d–i). Interestingly, activation of the double mutant by DMAEA, which differs from ACh by a single methyl group, is much less effective than observed with wild-type ELIC. This demonstrates that the two substitutions that abolish ACh antagonism, also impair agonism by this structurally related ACh derivative. Finally, the addition of a single methyl group, converting the tertiary amines, DMAEA and trimethylamine, into the quaternary amines, ACh and tetramethylammonium, renders both molecules ineffective, as they fail to activate either the A75D/F133W double mutant or wild-type ELIC (Fig. 4j–o).

To illustrate the differences in ligand sensitivity of wild-type ELIC and the A75D/F133W double mutant, we compared their responses to both ACh and GABA (Fig. 5). When 25 mM ACh is

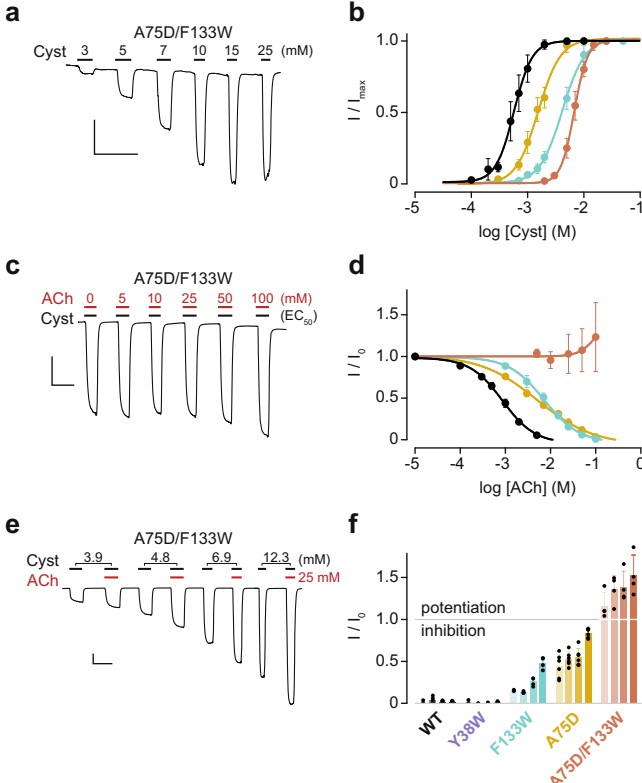

**Fig. 3 Cysteamine and ACh sensitivity of the A75D/F133W double mutant ELIC. a, c** Representative whole-cell traces and **b, d** corresponding dose response curves are shown for **a, b** cysteamine (Cyst) activation and **c, d** ACh inhibition of cysteamine-activated currents. In both cases, the duration of cysteamine and ACh applications (black and red bars, respectively) and ligand concentrations (mM) are indicated above the traces. In (**c**) the cysteamine concentration was fixed at its $EC_{50}$ for the A75D/F133W double mutant (6.9 mM). The dose response curves for wild-type ELIC (black), A75D (orange), F133W (green) and A75D/F133W (burgundy) are shown. Error bars in (**b, d**) represent one standard deviation from the mean, obtained from a minimum of 6 independent oocyte replicates. **e** The cysteamine concentration corresponding to $EC_{10}$ (3.9 mM), $EC_{20}$ (4.8 mM), $EC_{50}$ (6.9 mM), and $EC_{90}$ (12.3 mM) is indicated above each peak, with ACh fixed at 25 mM (red bars). **f** The normalized response at each cysteamine concentration for wild-type (WT) and various mutant ELICs are shown as a bar graph with each data point overlaid as a dot. Each mutant and the wild type were characterized with four cysteamine concentrations, corresponding to $EC_{10}$, $EC_{20}$, $EC_{50}$, and $EC_{90}$ (left to right) are depicted as different saturation levels (light to dark; $EC_{10}$ to $EC_{90}$). Error bars represent one standard deviation from the mean, obtained from a minimum of 4 independent oocyte replicates. The *x*- and *y*-axis of the scale bar in panels (**a, c, e**) corresponds to 1 min and 0.5 μA, respectively.

added to wild-type ELIC activated by a concentration of cysteamine corresponding to its $EC_{50}$, the cysteamine-activated current is completely inhibited (Fig. 5a). This demonstrates that ACh is an effective antagonist of cysteamine-activated currents in wild-type ELIC. When instead of ACh, 25 mM GABA is added to wild-type ELIC, the cysteamine-activated current is increased roughly 2-fold (Fig. 5b). GABA is a full agonist of wild-type ELIC, and at 25 mM (i.e. approximately 10-fold the GABA $EC_{50}$) it is expected to produce a near maximal response, which is twice as large as what is both predicted and observed when cysteamine is applied alone at a concentration corresponding to its $EC_{50}$. In stark contrast, for the A75D/F133W double mutant, 25 mM of

either ACh or GABA leads to the same small degree of potentiation (Fig. 5c, d), which can be explained by the aforementioned indirect effects of increased osmolarity and ionic strength. Thus, for the A75D/F133W double mutant both ligands have minimal effect on the cysteamine-activated currents. Evidently, these two substitutions are sufficient to eliminate both ACh antagonism and GABA agonism in ELIC.

**Acetylcholine sensitivity of ELIC harbouring the L9′A substitution.** We investigated whether ACh was trapping ELIC in a desensitized conformation without appreciably, or at least noticeably, activating it. This phenomenon has been observed with various ligands targeting neuronal AChRs[27–31], and is thought to underlie the mechanism of *d*-tubocurarine antagonism of the muscle-type AChR[32–34]. To test this hypothesis, we took advantage of the well-characterized L240A substitution (L9′A), which maps to the highly conserved M2 transmembrane region lining the ELIC pore. Substitutions at this 9′ position in ELIC and AChRs have been shown to dramatically slow agonist-induced desensitization and increase apparent agonist potency[35–39]. We evaluated whether the L9′A substitution would similarly slow desensitization and increase agonist potency for ELIC, thereby exposing any hidden ACh agonism. Consistent with previous data, the desensitization of the L9′A mutant was dramatically slowed in comparison to wild-type (Fig. 6a)[35,40], and at the same time the potency of cysteamine was increased (Fig. 6b, Supplementary Fig. 9a, Table 1). We also installed the L9′A substitution into the A75D/F133W double mutant (L9′A + A75D/F133W), which also led to an increase (albeit more modest) in the potency of cysteamine (Fig. 6b, Supplementary Fig. 9b, Table 1). We then tested whether ACh could elicit a response in either of these L9′A mutants. Neither mutant produced observable current when increasing concentrations of ACh were applied, but application of 5 mM cysteamine produced a robust peak current in both cases (Fig. 6c, d). In addition, both L9′A mutants showed similar sensitivity to ACh inhibition as their parent channels (Supplementary Fig. 10). Evidently, the presence of this L9′A substitution is insufficient to expose ACh agonism in either the wild-type ELIC or the A75D/F133W double mutant, suggesting that rapid desensitization upon ACh binding is not the origin for the apparent lack of ACh agonism.

## Discussion

To gain insight into the structural origins of agonism and antagonism in ELIC, we asked whether it was possible to convert ACh from an ELIC antagonist into an ELIC agonist by substituting AChR residues into ELIC. First, we confirmed that ACh inhibits agonist-activated currents through ELIC, inhibiting both cysteamine and GABA responses with a similar potency. To install ACh agonism, four candidate residues for mutation were identified using available structures and statistical coupling analysis. Substituting the corresponding muscle-type AChR residues at these four sites led to four single mutants that all had altered sensitivity to ACh as an antagonist, as well as cysteamine as an agonist. While none of the single substitutions converted ACh into an ELIC agonist, the A75D and F133W substitutions each decreased the sensitivity of ELIC to ACh inhibition. Combining these two substitutions to form the A75D/F133W double mutant abolished the antagonist activity of ACh.

Our hypothesis was that substituting AChR residues into the agonist-binding site of ELIC would allow ACh to both bind and stabilize ELIC in an open conformation, thereby converting ACh into an ELIC agonist. The hypothesis that subtle, local, structural changes would suffice to install ACh agonism seemed reasonable given that the ACh analogue DMAEA is able to activate ELIC[10]. At

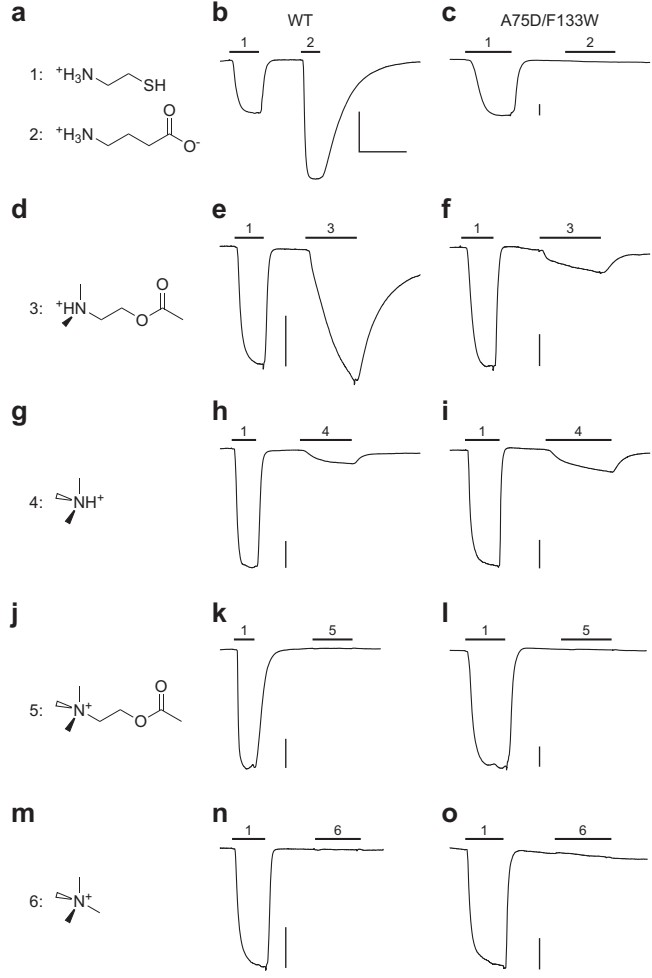

**Fig. 4 Activity of wild-type (WT) and the A75D/F133W double mutant ELIC in the presence of various amines.** Representative whole-cell responses for WT and the A75D/F133W double mutant ELIC are shown when 25 mM **a**–**c** GABA (compound 2), **d**–**f** 2-dimethylaminoethyl acetate (DMAEA; compound 3), **g**–**i** Trimethylamine (compound 4), **j**–**l** ACh (compound 5), or **m**–**o** Tetramethylammonium (compound 6) are added at the indicated times. In each case, an initial application of cysteamine at its respective $EC_{50}$ was added as a positive control (compound 1), while the second application is the response after application of 25 mM of the indicated amine (compounds 2–6). The vertical scale bar for each panel represents 0.5 μA, while the horizontal bar shown in panel **b** represents 1 min and applies to all traces.

the very least, we expected that introducing AChR residues, and making the ELIC agonist-binding site more AChR-like, would increase the affinity of ELIC for ACh. Thus, even if we failed to convert ACh into an agonist, we expected that the potency of ACh antagonism would increase, since higher affinity for ACh would make ACh more effective at outcompeting agonists. Of the substitutions tested, only the Y38W substitution led to an increase in the sensitivity of ELIC to ACh inhibition. Both the A75D and F133W substitutions decreased the sensitivity of ELIC to ACh, and when combined, abolished ACh antagonism altogether. While we cannot exclude the possibility that ACh still binds to the A75D/F133W double mutant, given that ACh no longer competes with cysteamine in this mutant, the simplest interpretation is that ACh no longer binds to the agonist-binding site, and that together these two mutations effectively eliminate ACh binding.

On their own, the A75D and F133W substitutions decreased the potency of ACh inhibition by 7-fold and 8-fold, respectively. If these residues contributed independently to ACh inhibition, then combining them would have an additive effect, resulting in an approximately 60-fold decrease in ACh sensitivity of the double mutant (and an expected increase in the ACh $IC_{50}$ to approximately 60 mM). Instead, no inhibition by ACh is observed, even when 100 mM ACh is present. This result demonstrates that these two residues have a synergistic effect on ACh antagonism. In several AChR[16,33,41–43] and AChBP[20,21] structures, the corresponding aspartic acid is close enough to form a hydrogen bond with the backbone N-H of the tryptophan, and this interaction is thought to be important for optimally positioning the tryptophan for high affinity ACh binding[26]. In the AChR and AChBP the functional dependence between these two residues can be explained by this physical interaction. For the double mutant ELIC, the functional dependence of these two residues suggests that a physical interaction remains, however with a different outcome – affinity for ACh is apparently abolished. Thus, residues deemed important for ACh binding in one background unexpectedly abolish it in another. These results not only demonstrate that additional substitutions are required to convert ACh into an ELIC agonist, but perhaps more importantly that the background upon which residues exist determines their specific contribution to function. This suggests that agonist specificity in pentameric ligand-gated ion channels is highly contingent upon evolutionary history, similar to what has been observed with glucocorticoid receptors[44,45].

Statistical coupling analysis has been used to uncover structure-function relationships in a variety of ion channels[46–49]. Despite evidence that statistically coupled positions overlap with functionally important sites within AChRs[50], the method has been relatively unexploited for pentameric ligand-gated ion channels. With the ever-growing database of sequences, and the increasing collection of pentameric ligand-gated ion channel structures, statistical coupling analysis is poised to uncover structure-function relationships obscured by epistasis. For example, our results reinforce the notion that although agonist action stems from local interactions between agonists and the amino acid residues within their binding sites, the mechanisms by which agonists elicit their responses is a global property of the protein, dependent upon cryptic contributions from distant residues. Indeed, it is well documented that a large number of disease-associated mutations map to residues distal from the agonist-binding site[51–54]. Many of these distal mutations influence diverse aspects of AChR function including: agonist affinity[51], gating kinetics[55,56], and both inter-subunit[54] and intra-subunit allosteric communication[55,56]. In addition, studies of chimeric channels where the extracellular domain of one channel is combined with the transmembrane domain of another have repeatedly shown that residues at the domain interface influence agonist activity in a nontrivial manner[25,57,58]. In particular, in an ELIC/α7 chimera, the potency of cysteamine was influenced by residues at the interface between the ELIC extracellular domain and the α7 transmembrane domain[58].

Our AChBP and α7-based statistical coupling analyses identified 43 and 89 statistically coupled sites, respectively (Fig. 2; Supplementary Fig. 2; Table 2). Of the large number of residues that differ between AChBP or α7 and ELIC at these sites, we have only substituted four, with all four being close to, or in contact with, bound agonist. A possible shortcoming of focusing on these four residues is that some (or all) of the unchanged residues remain incompatible with ACh agonism. By restricting our mutagenesis to residues within the agonist-binding site we have not fully exploited the ability of statistical coupling analysis to detect long-range epistatic interactions between residues. To re-wire ELIC, and convert ACh into an agonist, it may be necessary to transplant an entire sector from a related AChR into ELIC.

A limitation of our structure-based approach is that it only identifies statistically coupled sites encompassed within the utilized

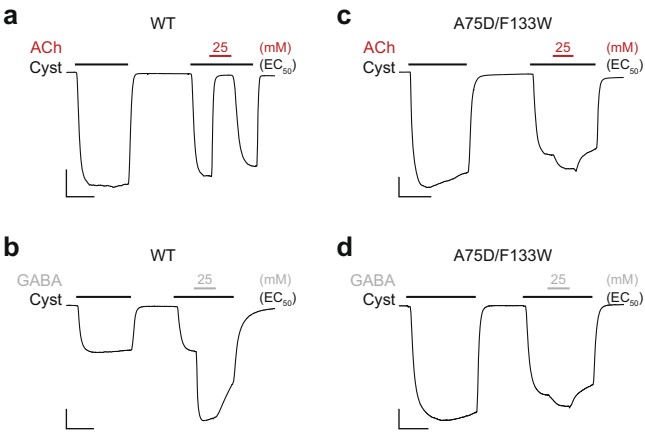

**Fig. 5 Contrasting effects of ACh and GABA on wild-type versus the A75D/F133W double mutant ELIC.** Representative whole-cell traces for **a**, **b** wild-type (WT) and **c**, **d** the A75D/F133W double mutant. In each case, the first peak shows the response to a 2 min application of cysteamine at its corresponding $EC_{50}$ (black bar). The second peak shows the response to another 2 min application of cysteamine, again at its $EC_{50}$, but which was interrupted by a 40 s pulse where 25 mM ACh red bar; (**a**, **c**) or 25 mM GABA gray bar; (**b**, **d**) was co-applied. The x- and y-axis of the scale bar in each panel corresponds to 1 min and 0.5 μA, respectively.

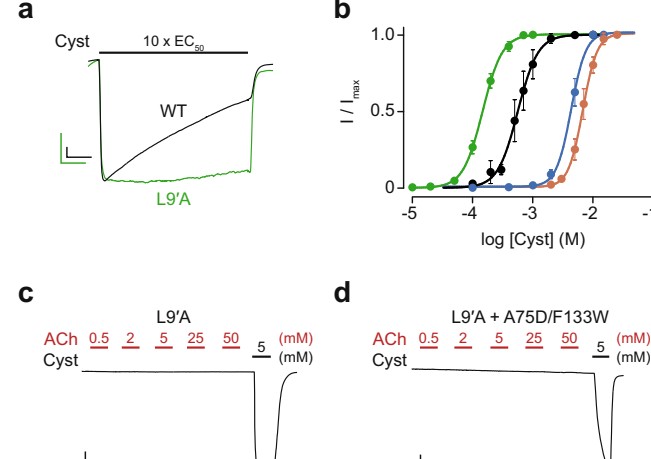

**Fig. 6 Rapid desensitization does not explain the inability of ACh to activate ELIC.** **a** Representative whole-cell traces of the L9′A mutant (green) relative to wild-type (WT) ELIC (black). A saturating concentration of cysteamine (10 x $EC_{50}$) was applied to each oocyte for 6 min ($n = 3$ and 4 independent oocytes for WT and L9′A, respectively). Maximum current amplitudes were normalized and overlaid to facilitate qualitative comparison of the macroscopic desensitization rates. **b** Dose-response curves of cysteamine activated WT ELIC (black), the L9′A mutant (green), the A75D/F133W double mutant (burgundy), and the L9′A + A75D/F133W triple mutant (blue). A minimum of 6 independent oocyte replicates were performed for each mutant. Error bars represent one standard deviation from the mean. **c**, **d** No ACh-induced currents were observed for either **c** the L9′A mutant or **d** the L9′A + A75D/F133W triple mutant ELIC upon 30 s applications of increasing concentrations of ACh (0.5 to 50 mM), whereas a 5 mM application of cysteamine led to robust activation in both cases. A total of seven and four independent oocyte replicates were collected for the L9′A and the L9′A + A75D/F133W mutants, respectively. The x- and y-axis of the scale bars in (**a**, **c**, **d**) correspond to 1 min and 0.5 μA, respectively.

template structures. This is most obvious for the AChBP-based analysis, which by design cannot detect statistically coupled sites in the ELIC transmembrane and cytoplasmic domains, given that AChBP lacks both. Similarly, 167 residues comprising much of the cytoplasmic domain of the human α7 AChR are not modelled within the structural template used in our α7-based analysis (PDB ID: 7KOX), and thus statistically coupled residues in this region also cannot be detected. Although ELIC has a small cytoplasmic domain that does not share homology with α7, the cytoplasmic domain of eukaryotic pentameric ligand-gated ion channels has been shown to affect agonism[59]. To identify statistically coupled sites within this region it will be necessary to exploit a recent structure of the α7 cytoplasmic domain[60], or perform a traditional statistical coupling analysis based purely on sequence similarity as opposed to the structure-based approach imposed in the present analysis.

Finally, given the difficulties with stabilizing biologically relevant conformations of ELIC for crystallization[35], we cannot exclude the possibility that ACh stabilizes a desensitized state that is not faithfully represented in the ACh-bound ELIC crystal structure. Nevertheless, considering our data with the L9′A mutants, and working under the assumption that the ACh-bound ELIC crystal structure is biologically relevant, we propose that ACh antagonism of ELIC can be viewed as an extreme case of partial agonism. At first glance, the concept of agonism seems relatively simple: agonists bind to their receptors and activate them. However, it is important to remember that not all agonists are equal. Partial agonists are less effective than full agonists at activating their receptor. To explain partial agonism, del Castillo and Katz proposed that receptor activation could be broken down into a two-step process, where in the first step the agonist binds to an inactive receptor, that in the second step switches to an active conformation[61]. Full versus partial agonists differ in their ability to bring about the second/conformational change step, with full agonists being more effective than partial agonists[62]. This two-step process reveals two properties inherent to agonists: affinity and efficacy[63]. To be effective, agonists must both (1) bind to their receptor (affinity), and (2) stabilize the active conformation of the receptor (efficacy). Simply binding with high affinity is not sufficient for agonism. The interaction between ACh and ELIC is a good case

in point. ACh binds to ELIC, bringing it to the verge of activation[10], but ultimately fails to stabilize ELIC in an active conformation. This is analogous to what has been observed in structures of the related serotonin receptor in complex with several setron antagonists, which appear to stabilize intermediates along the activation pathway[64,65]. ACh may be similarly stabilizing ELIC in an intermediate along its activation pathway. These phenomena reflect a relatively under-appreciated mechanism of competitive antagonism, in which agonists with appreciable affinity, but negligible efficacy, present as competitive antagonists. Although the result is the same, this mechanism of competitive antagonism contrasts with that of classic AChR antagonists, such as *d*-tubocurarine and α-bungarotoxin, that either trap the channel in a desensitized state[33], or arrest it in an antagonized/closed conformation resembling the resting state[22,41,66].

## Methods

**Materials.** Cysteamine, acetylcholine chloride, γ-aminobutyric acid, and atropine were purchased from Sigma-Aldrich. Dithiothreitol, 2-dimethylaminoethyl acetate, tetramethylammonium chloride, and trimethylamine hydrochloride were purchased from Fisher Scientific.

**Homologous sequence searches and multiple sequence alignments.** We performed two separate Statistical Coupling Analyses, each requiring its own multiple sequence alignment. As described[15,17], sequence profiles based upon a structure of (1) the *Lymnaea stagnalis* acetylcholine binding protein (AChBP) in complex with ACh (PDB ID: 3WIP)[11], and (2) the human α7 AChR in complex with epibatidine and PNU-120596 (PDB ID: 7KOX)[16], were used to search for homologous protein sequences that presumably adopt the same backbone structure as their respective template. Briefly, using one of the AChBP or α7 subunits (Chain A from PDB ID:

3WIP; Chain A from PDB ID: 7KOX) as a template, a Hidden Markov Model was built with the *hmmbuild* tool (HMMER 3.2.1 suite) from 150 new sequences proposed by the *fixbb* tool within Rosetta 3.9 software suite[67]. Each Hidden Markov Model was used to query the UniRef100 protein sequence database (AChBP: June 2019; α7: March 2022) using *hmmsearch*. These searches yielded 7132 and 32,895 sequences for the AChBP and α7 templates, respectively, each with E-values less than or equal to 0.01. In each case sequences were locally aligned to their respective Hidden Markov Model using *hmmalign*. To reduce redundancy, sequences were grouped into 90% identity clusters using the program Cluster Database at High Identity with Tolerance (CD-HIT)[68]. Sequences with more than 90% identity to any other sequence within each cluster were removed, yielding final multiple sequence alignments of 1438 and 5020 representative sequences for AChBP and α7, respectively, where no two sequences in each alignment were more than 90% identical to each other.

**Statistical coupling analysis (SCA).** The SCA v5.0 toolbox[69] was used to analyze the final multiple sequence alignments ignoring positions with more than 40% gaps (i.e., gap cut-off of 0.4), which in the case of α7 resulted in exclusion of the MA, MX, and M4 helices, as well as the cytoplasmic domain. In each case, the analysis suggested the presence of multiple protein sectors, and so the statistically significant top 4 eigenmodes (i.e. kmax = 4) of the positional correlation matrix were transformed into maximally independent components through independent component analysis[12]. The top three independent components formed the basis of three independent protein sectors, where a cut-off of 0.95 was used to define the sites included in each sector[12]. The corresponding sector residue positions from AChBP (PDB ID: 3WIP)[11] and α7 (PDB ID: 7KOX)[16] were mapped onto ELIC (PDB ID: 3RQW)[10].

**Molecular biology and mRNA expression.** The ELIC-pTLN plasmid containing the wild-type ELIC gene was provided by Dr. Raimund Dutzler[70]. A C-terminal alanine, which is a cloning artefact absent in the GenBank sequence (GenBank/UniProt: P0C7B7), was removed. Individual substitutions were introduced by inverse PCR[71], and confirmed by DNA sequencing. The double mutant construct (A75D/F133W) was obtained from Twist Bioscience. The ELIC-pTLN constructs were linearized with *Mlu*I and transcribed from the SP6 promoter region. Capped ELIC complementary RNA (cRNA) was produced by in vitro transcription using the mMESSAGE mMACHINE SP6 kit (Ambion). cRNA concentration was determined using its absorbance at 260 nm, and its integrity was determined by agarose gel electrophoresis.

**Electrophysiology.** Stage V-VI *Xenopus laevis* oocytes were isolated[72] and deflocculated enzymatically in ND96 buffer without $Ca^{2+}$ (96 mM NaCl, 2 mM KCl, 1 mM $MgCl_2$, 2 mM pyruvate, 5 mM HEPES, pH 7.4), but supplemented with 1 mg/mL collagenase B and 1 mg/mL trypsin inhibitor, for 2 h at room temperature under constant stirring. The deflocculation buffer was then gradually removed by washing the oocytes in ND96 buffer (96 mM NaCl, 2 mM KCl, 1 mM $MgCl_2$, 1 mM $CaCl_2$, 2 mM pyruvate, 5 mM HEPES, pH 7.4) for several iterations. The oocytes were then allowed to stir for an additional hour at room temperature under constant stirring with the buffer being periodically replaced until clear. To obtain optimal receptor expression, individually selected and healthy oocytes were injected with 0.2 ng cRNA for wild-type ELIC and 5 ng cRNA for mutants. Subsequently, oocytes were incubated for one day at 16 °C in ND96+ buffer (96 mM NaCl, 2 mM KCl, 1 mM $MgCl_2$, 1 mM $CaCl_2$, 2 mM pyruvate, and 5 mM HEPES at pH 7.6). After expression, injected oocytes were placed in a RC-1Z oocyte chamber (Harvard Apparatus; Hamden, CT) containing buffer (pH 7.4) composed of 150 mM NaCl, 0.5 mM $BaCl_2$, 10 mM HEPES, and 1 mM dithiothreitol or 1 μM atropine if cysteamine, or ACh, was used in the experiment, respectively. The presence of the reducing agent, dithiothreitol, ensures that the sulfhydryl group of cysteamine remains reduced. Atropine, a competitive antagonist to endogenous muscarinic AChRs, prevents activation of endogenous channels such as calcium-activated chloride channels via muscarinic AChRs[73,74].

Once in the oocyte chamber, whole-cell currents were recorded using a two-electrode voltage-clamp apparatus (OC-725C oocyte clamp; Harvard Apparatus), with a buffer flow rate of 5–10 mL min$^{-1}$. Currents in response to various additions of cysteamine and/or ACh, were measured with the transmembrane voltage clamped at −60 mV. Dose response curves for each wild-type and mutant ELIC were acquired from at least five oocytes in two different batches.

**Data analysis.** For identifying the half maximal effective concentration ($EC_{50}$) and apparent Hill coefficient ($n_H$) of cysteamine, dose response curves were created by taking the magnitude of each cysteamine-activated peak current (I) and normalizing this value to the maximal peak current in the presence of excess agonist ($I_{max}$). The log (base 10) cysteamine concentration in molar was then plotted on the x-axis against the fractional response ($I/I_{max}$) on the y-axis. The data was fitted using nonlinear regression with a variable slope sigmoidal dose response (Eq. 1, below) in GraphPad Prism (v.8.0.0).

$$y = \frac{1}{1 + 10^{n_H(LogEC_{50} - X)}} \quad (1)$$

Individual $EC_{50}$ and Hill coefficients were averaged for each oocyte to derive a mean ± standard deviation. For identifying a half maximal inhibitory concentration

($IC_{50}$) and Hill coefficient of ACh, a concentration of cysteamine corresponding to its $EC_{50}$ was used in the presence of increasing concentrations of ACh. Peak currents elicited from cysteamine with ACh (I) were normalized to the initial cysteamine peak current in the absence of ACh ($I_o$). The log (base 10) ACh concentration was then plotted on the x-axis against the fractional response ($I/I_o$) on the y-axis. Subsequent steps were the same as above for deriving a cysteamine $EC_{50}$ and Hill coefficient. The individual $IC_{50}$ and Hill coefficients were averaged for each oocyte to derive a mean ± one standard deviation. Lastly, to identify four normalized response values, cysteamine at the 10% maximal effective concentration ($EC_{10}$), 20% maximal effective concentration ($EC_{20}$), $EC_{50}$ and 90% maximal effective concentration ($EC_{90}$), was applied in the presence and absence of 25 mM ACh. The cysteamine-activated peak current with 25 mM ACh, was normalized to the cysteamine-activated peak current without ACh.

**Statistics and reproducibility.** We defined a replicate as a separate oocyte injected with the same cRNA. Dose response curves for each wild-type and mutant ELIC were acquired from at least five oocytes, with a minimum of two oocytes derived from a different batch. Statistical tests were performed using GraphPad Prism (v.8.0.0). All log($EC_{50}$) comparisons and log($IC_{50}$) comparisons were done using a one-way ANOVA followed by Dunnett's post-hoc test relative to cysteamine activated wild-type ELIC. We used the Dunnett's multiple comparison test to avoid an inflated Type I error rate. All significant findings had an adjusted p-value less than the alpha level of 0.001. For our co-application experiments we compared the level of potentiation produced by ACh to the three other conditions using a one-way ANOVA followed by Dunnett's post-hoc test. All significant findings had an adjusted p-value less than the alpha level of 0.05. Summaries of all statistical tests can be found in Supplementary Tables 1–3.

**Reporting summary.** Further information on research design is available in the Nature Portfolio Reporting Summary linked to this article.

## Data availability

All source data associated with the current study are available in a figshare repository (https://doi.org/10.6084/m9.figshare.19127057.v3). This includes: UniRef100 accession codes for all sequences used in statistical coupling analyses, multiple sequence alignment files, and whole-cell two-electrode voltage clamp traces and analyzed data. All other data are available upon request.

## Code availability

MATLAB scripts to run the statistical coupling analyses associated with the current study are freely available on figshare (https://doi.org/10.6084/m9.figshare.19127057.v3).

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

## Acknowledgements

M.S. was the recipient of a Natural Sciences and Engineering Research Council (NSERC) of Canada CREATE Scholarship. R.A.C. acknowledges grants from the Natural Sciences and Engineering Research Council of Canada (RGPIN-2016-04831), and the Canada Foundation for Innovation (26503). C.J.B.d.C. acknowledges grants from the Natural Sciences and Engineering Research Council of Canada (RGPIN-2016-04801), the Canada Foundation for Innovation (34475), and the Canadian Institutes of Health Research (377068). This work was also funded by a New Frontiers in Research Fund – Exploration Grant (NFRFE-2018-00064) awarded to C.J.B.d.C. and R.A.C.

## Author contributions

M.S. and M.J.T. acquired and analyzed all electrophysiology data. J.A.B.V. and R.A.D. performed statistical coupling analysis, J.A.B.V. identified sites for mutagenesis. J.A.B.V., M.J.T. and M.S. prepared the DNA constructs. M.J.T. and J.E.B. provided the *Xenopus* oocytes, two-electrode voltage-clamp apparatus, and assisted in technical aspects of the electrophysiology experiments. C.J.B.d.C, R.A.C, J.A.B.V, M.J.T., and M.S. designed the experiments and interpreted the data. C.J.B.d.C, J.A.B.V and M.S. wrote the manuscript. C.J.B.d.C and R.A.C supervised the project. All authors were involved in manuscript editing.

## Competing interests

The authors declare no competing interests.
