## [Peer Review File · Communications Biology]

Reviewers' comments:

Reviewer #1 (Remarks to the Author):

In this manuscript, the authors investigated structural origins of agonism/antagonism in ELIC. Their original hypothesis is that substituting AChR residues into the agonist-binding site of ELIC would allow ACh to both bind and stabilize ELIC in an open conformation, thereby converting ACh into an ELIC agonist. To test this hypothesis, they generated several ELIC mutants and performed electrophysiology functional measurements on these mutants. They found ACh remained as an antagonist with varied potencies in all of their ELIC mutants except the double mutant A75D/F133W, in which ACh abolished its antagonist activity and was able to potentiate channel currents at a high concentration (20 mM). While the study results do not positively support their initial hypothesis, they are valuable in helping to understand what define agonism/antagonism in ELIC or in ligand gated channels in general. Overall, the experiments were conducted carefully, the results are well organized, and the discussion is balanced. A few points, as listed below, need to be addressed.

1. It is adequate to use the structure of an ACh-bound AChBP as a template in the design of ELIC mutations, including in statistical coupling analysis (SCA). However, it is necessary to perform a similar SCA on one of the currently available AChR structures. Residues of ELIC vs. AChR at the interface of the extracellular domain (ECD) / transmembrane domain (TMD) and in the TMD identified by SCA may offer clues as to why the attempt to convert ACh from an antagonist to an agonist failed.
2. The authors raised the notion that "... agonists elicit their responses is a global property of the protein, dependent upon cryptic contributions from distant residues." The notion is important, but the supporting evidence for this notion is merely a citation of a SCA publication. The authors should elaborate it through previous experimental results, particularly those closely related findings. See #3.
3. Several previous studies showed that an ELIC agonist can have very different potencies in chimeric channels, which have the identical ELIC extracellular agonist binding domain but varied transmembrane domains. Ligand binding is only one of the many parts in channel activation. Whether a ligand serves as an agonist depends not only on how tightly it binds to the orthosteric binding sites, but also on how easily the binding signal can be transduced for channel opening. Similar experimental supports may also be found from chimeric AChBP channels. ACh activates the chimeric AChBP-5HT3A receptor (ref.32) as described in the manuscript, but ACh may not be able to activate other chimeric channels that have the same AChBP in the ECD but different transmembrane domains.
4. It is possible that ACh retains the same binding affinity in the A75D/F133W mutant, but the double mutations reduced "resistance" for channel opening so that the same ACh binding became capable of contributing to stabilizing an open-channel conformation. The authors should check this possibility through in-silico mutations on available ELIC structures and discuss such a possibility.
5. A paragraph of discussion, highlighting limitations of the current study and potential future studies to further explore structural origins of agonism/antagonism will be helpful to readers.

Reviewer #2 (Remarks to the Author):

Here Slobodyanyuk et al. present a structure-function study of the pLGIC ELIC. They seek to use structure-based statistical analysis to find a way to mutate ELIC such that ACh switches from being an antagonist to being an agonist. They are ultimately unsuccessful- mutations result in a loss rather than a gain of ACh activity. The conclusion is that regions outside the orthosteric agonist site must be involved in determination of agonist vs. antagonist activity. Overall, the study is clearly written and appears to be carefully performed. The investigation and resolution of why ACh causes potentiation at high doses is satisfying. As currently carried out the study is interesting to me, someone very close to the field, but would be much more interesting if a robust mechanistic finding had been uncovered, rather than just losing ACh binding. As currently presented the reader does not learn much about ELIC or other pLGICs. Negative findings that are well supported are still useful and informative, and I think

these are or could be. To increase reader interest, I have the following suggestions for consideration; most are minor text edits.

1. Beginning in Introduction, states are referred to as closed and open. These terms denote conformations, not states; there are multiple closed conformations and perhaps even multiple open conformations. I suggest using resting, activated, and desensitized terminology to describe states, and closed and open to describe conformations.
2. End of the introduction. I agree that this is an underappreciated way of stabilizing a non-conducting conformation- essentially through stabilizing a desensitized state without first heavily populating an activated state. There are examples, though, and it might be worth mentioning these. At nAChRs, sazetidine A (PMID: 16857741), AT-1001 (PMID: 25180076, 31488329) are a couple. d-Tubocurarine also likely works this way (very low efficacy agonist = functional antagonist). Could mention in discussion if it fits better there and you agree.
3. If ELIC desensitizes, which the literature suggests it does, ACh maybe acting like the ligands mentioned above, and stabilize a high affinity agonist-bound desensitized-like state. 9' mutations in the pore that alter desensitization have been effective at making it easier for fast desensitizing channels to stay open or, put another way, agonists that cause rapid desensitization (for example in $\alpha 7$) to be more efficacious. A mutant like this might be required to 'see' ACh agonist activity.
4. Paragraph starting line 226. Please consider toning down the enthusiasm for AChBPs as being useful in probing tertiary/quaternary conformational changes associated with state transitions. To my knowledge, the only parts that move upon agonist binding, based on the available structures, are loop C and to some degree loop F- regions that directly participate in orthosteric ligand binding. Perhaps when coupled to a channel, with mutations in the linkage regions, AChBP can promote gating, but it is not very good at it. To be clear, those soluble receptor homologs are spectacularly useful, but they are not good tools for studying allostery.
5. The final section of the discussion ignores desensitization- if one wants to talk about functional efficacy, a two state (resting vs. activated) model is too simple for pLGICs. I could be completely wrong in my idea that perhaps ACh is stabilizing a desensitized state in a real membrane, but the possibility seems worth considering. The bacterial pLGICs are notoriously difficult to get to adopt different conformations for structural biology purposes, especially ELIC- so the channel conformation may not be perfectly informative.

Reviewer #3 (Remarks to the Author):

The title of the manuscript "Origins of acetylcholine antagonism in a bacterial pentameric ligand-gated ion channel" in general seems to be correct and justified. To my mind, I would use the word "origin" and would add the restricting abbreviation "ELIC" because acetylcholine is an antagonist of this particular representative of bacterial Cys-loop receptors.

Although the title claims an interest in antagonism, in fact the main task of authors was to modify the ELIC in such a way that acetylcholine would start functioning as an agonist. In spite of the large experimental work this task was not solved and in this way the authors are correct not placing the word "agonist" to the title. The main result that the authors demonstrated is that ELIC mutants A75D and F133W needed considerably higher concentrations to inhibit binding of agonists (thus indicated the points of attachment of the acetylcholine to its antagonist-binding site, and in a way localizing such a sight). However, a double mutant A75D/F133 did not behave as an antagonist at all, not precluding binding of cysteamine, an ELIC agonist – and no explanation of this fact could be provided. AChBP is an excellent model for nicotinic receptors and all other Cys-loop receptors and the authors started with the X-ray structure of the AChBP complex with acetylcholine to choose those mutations which would ensure the ACh agonistic activity on ELIC. As mentioned, it did not work. We should not forget that the ACh binding to AChBP does not produce a current (no channel is available), the binding sites of agonists and antagonists are considerably overlapping, and the main difference in those X-ray structures between complexes with agonists and antagonists is the movement of loop C to the center for the former and for the latter to the periphery. May be the authors could have got better hints for

mutations in the ELIC not from the structure of AChBP, but from the X-ray and cryo-EM structures of the LBD complexes with agonists (not compulsory with acetylcholine, for example alpha7-AChBP chimera in complex with epibatidine, alpha4beta2 nAChR with bound epibatidine, recent structure of the alpha7 receptor in the activated state).

I was surprised that the authors did not mention that acetylcholine is also an antagonist of the alpha9alpha10 nicotinic receptors and some, not very successful, attempts were made to detect the residues in the binding sites of the alpha9 subunit responsible for this (although information on the residues involved in binding of alpha-neurotoxins and various alpha-conotoxins both on the AChBP and various nAChR subtypes is available).

In the manuscript the authors demonstrated the important role of residues which in the X-ray structure of the AChBP complex with ACh appear quite distant from the bound ACh. Similar conclusions have been also done earlier -see, for example, Inter-residue coupling contributes to high-affinity subtype-selective binding of α -bungarotoxin to nicotinic receptors. Sine SM, Huang S, Li SX, daCosta CJ, Chen L. *Biochem J.* 2013 Sep 1;454(2):311-21

I would appreciate if the authors formulate what is the main conclusion of this manuscript and what is its novelty for the whole family of the Cys-loop receptors

Minor comment

No need to introduce to the Abstract "conformational changes indicative of activation"- it was a sort of a hypothesis formulated by other authors.

Response to Review

Reviewers / Editor

Authors

Revised text

Editorial Summary:

We therefore invite you to revise and resubmit your manuscript, taking into account all the comments of the reviewers. In particular, reviewer #1 has indicated that it is necessary to perform a similar SCA on one of the currently available AChR structures. Additionally, all of the reviewers have indicated that they would like to see more insight into the significance of your results.

The editor points out two main things to be addressed in this revision:

1. "...it is necessary to perform a similar SCA on one of the currently available AChR structures."

At the request of the Reviewer #1 we have performed an additional SCA based upon the 2021 structure of the human $\alpha 7$ acetylcholine receptor in complex with the agonist epibatidine and the positive allosteric modulator PNU-120596 (PDB: 7K0X; PMID: 33735609).

2. "...more insight into the significance of your results."

We agree and have emphasized the significance of our results in the Abstract, Introduction, and Discussion. Here we highlight some specific changes in the Discussion below:

Lines 449 – 455 of our revised manuscript:

Thus, residues deemed important for ACh binding in one background unexpectedly abolish it in another. These results not only demonstrate that additional substitutions are required to convert ACh into an ELIC agonist, but perhaps more importantly that the background upon which residues exist determines their specific contribution to function. This suggests that agonist specificity in pentameric ligand-gated ion channels is highly contingent upon evolutionary history, similar to what has been observed with glucocorticoid receptors^{52,53}.

And,

Lines 519 – 525 of our revised manuscript:

These phenomena reflect a relatively underappreciated mechanism of competitive antagonism, in which *agonists* with appreciable affinity, but negligible efficacy, present as competitive antagonists. Although the result is the same, this mechanism of competitive antagonism contrasts with that of classic AChR antagonists, such as *d*-tubocurarine and α -bungarotoxin, that

either trap the channel in a desensitized state⁴¹, or arrest it in an antagonized/closed conformation resembling the resting state^{30,49,74}.

Reviewer #1 (Remarks to the Author):

In this manuscript, the authors investigated structural origins of agonism/antagonism in ELIC. Their original hypothesis is that substituting AChR residues into the agonist-binding site of ELIC would allow ACh to both bind and stabilize ELIC in an open conformation, thereby converting ACh into an ELIC agonist. To test this hypothesis, they generated several ELIC mutants and performed electrophysiology functional measurements on these mutants. They found ACh remained as an antagonist with varied potencies in all of their ELIC mutants except the double mutant A75D/F133W, in which ACh abolished its antagonist activity and was able to potentiate channel currents at a high concentration (20 mM). While the study results do not positively support their initial hypothesis, they are valuable in helping to understand what define agonism/antagonism in ELIC or in ligand gated channels in general. Overall, the experiments were conducted carefully, the results are well organized, and the discussion is balanced. A few points, as listed below, need to be addressed.

1. It is adequate to use the structure of an ACh-bound AChBP as a template in the design of ELIC mutations, including in statistical coupling analysis (SCA). However, it is necessary to perform a similar SCA on one of the currently available AChR structures. Residues of ELIC vs. AChR at the interface of the extracellular domain (ECD) / transmembrane domain (TMD) and in the TMD identified by SCA may offer clues as to why the attempt to convert ACh from an antagonist to an agonist failed.

We agree. Our revised manuscript includes a new statistical coupling analysis based upon the structure of the human $\alpha 7$ acetylcholine receptor in complex with epibatidine and PNU-120596 (PDB ID: 7KOX). Gratifyingly, this $\alpha 7$ -based analysis identifies the same four sites that were identified in our original AChBP-based statistical coupling analysis, and which informed our mutagenesis experiments.

2. The authors raised the notion that "... agonists elicit their responses is a global property of the protein, dependent upon cryptic contributions from distant residues." The notion is important, but the supporting evidence for this notion is merely a citation of a SCA publication. The authors should elaborate it through previous experimental results, particularly those closely related findings. See #3.

We agree. In our revised manuscript we explicitly elaborate upon this point and cite closely related findings as shown below:

Lines 464 – 474 of our revised manuscript:

...agonists elicit their responses is a global property of the protein, dependent upon cryptic contributions from distant residues. Indeed, it is well documented that a large number of disease-associated mutations map to residues distal from the agonist-binding site^{59–62}. Many of these distal mutations influence diverse aspects of AChR function including: agonist affinity⁵⁹, gating kinetics^{63,64}, and both inter-subunit⁶² and intra-subunit allosteric communication^{63,64}. In addition, studies of chimeric channels where the extracellular domain of one channel is combined with the transmembrane domain of another have repeatedly shown that residues at the domain interface influence agonist activity in a nontrivial manner^{33,65,66}. In particular, in an ELIC/ $\alpha 7$ chimera, the potency of cysteamine was influenced by residues at the interface between the ELIC extracellular domain and the $\alpha 7$ transmembrane domain⁶⁶.

3. Several previous studies showed that an ELIC agonist can have very different potencies in chimeric channels, which have the identical ELIC extracellular agonist binding domain but varied transmembrane domains. Ligand binding is only one of the many parts in channel activation. Whether a ligand serves as an agonist depends not only on how tightly it binds to the orthosteric binding sites, but also on how easily the binding signal can be transduced for channel opening. Similar experimental supports may also be found from chimeric AChBP channels. ACh activates the chimeric AChBP-5HT3A receptor (ref.32) as described in the manuscript, but ACh may not be able to activate other chimeric channels that have the same AChBP in the ECD but different transmembrane domains.

We agree. This comment speaks to the observation that various ECD-TMD chimeras formed from different channels (including ELIC) usually require extensive engineering of their ECD-TMD interface, and furthermore that the resulting chimeras can have a different functional profile from either of their parent channels.

In addition to addressing this (in part) in our response to Comment #2 above, our revised manuscript now also raises this potential limitation, and specifically how it relates to the AChBP as model for acetylcholine receptor agonism. In the pasted text below, we then use this potential limitation as justification for our new/additional statistical coupling analysis based upon the recent $\alpha 7$ acetylcholine receptor structure (PDB: 7K0X), which was not available when we performed our original statistical coupling analysis.

Lines 253 – 257 of our revised manuscript:

Nevertheless, AChBPs lack both transmembrane and cytoplasmic domains, and although a chimeric construct in which the AChBP from *Lymnaea stagnalis* was coupled to a 5-HT_{3A} receptor pore could be activated by ACh, it had to be extensively engineered³³, indicating that elements necessary for translating ACh-binding into channel activation might not be fully preserved in AChBPs.

4. It is possible that ACh retains the same binding affinity in the A75D/F133W mutant, but the double mutations reduced “resistance” for channel opening so that the same ACh

binding became capable of contributing to stabilizing an open-channel conformation. The authors should check this possibility through in-silico mutations on available ELIC structures and discuss such a possibility.

We agree that it is possible that ACh still binds to the A75D/F133W double mutant. To make it explicitly clear that we cannot rule out this possibility, in the revised manuscript we preface our favoured interpretation with:

Lines 432 – 435 of our revised manuscript:

While we cannot exclude the possibility that ACh still binds to the A75D/F133W double mutant, given that ACh no longer competes with cysteamine in this mutant, the simplest interpretation is that ACh no longer binds to the agonist-binding site, and that together these two mutations effectively eliminate ACh binding.

As for checking whether ACh still binds to the mutant through “in-silico mutations on available ELIC structures”, it is unclear how in-silico mutations would resolve the issue of whether or not ACh still binds to the double mutant. Strictly speaking this would require experiments directly measuring ACh binding. In any case, detailed computational studies aimed at providing meaningful insight in this area are beyond the scope of the present work.

5. A paragraph of discussion, highlighting limitations of the current study and potential future studies to further explore structural origins of agonism/antagonism will be helpful to readers.

We agree and have elaborated upon the limitations of the current study, and also suggested some future ways to overcome these limitations.

Lines 475 – 497 of our revised manuscript:

Our AChBP and $\alpha 7$ -based statistical coupling analyses identified 43 and 89 statistically coupled sites, respectively (Fig. 2; Fig. S2; Table 2). Of the large number of residues that differ between AChBP or $\alpha 7$ and ELIC at these sites, we have only substituted four, with all four being close to, or in contact with, bound agonist. A possible shortcoming of focusing on these four residues is that some (or all) of the unchanged residues remain incompatible with ACh agonism. By restricting our mutagenesis to residues within the agonist-binding site we have not fully exploited the ability of statistical coupling analysis to detect long-range epistatic interactions between residues. To re-wire ELIC, and convert ACh into an agonist, it may be necessary to transplant an entire sector from a related AChR into ELIC.

A limitation of our structure-based approach is that it only identifies statistically coupled sites encompassed within the utilized template structures. This is most obvious for the AChBP-based analysis, which by design cannot detect statistically coupled sites in the ELIC transmembrane and cytoplasmic domains, given that AChBP lacks both. Similarly, 167 residues comprising much of the cytoplasmic domain of the human $\alpha 7$ AChR are not modelled within the structural template used in our $\alpha 7$ -based analysis (PDB ID: 7K0X), and thus statistically coupled residues in this region also cannot be detected. Although ELIC has a small cytoplasmic domain

that does not share homology with $\alpha 7$, the cytoplasmic domain of eukaryotic pentameric ligand-gated ion channels has been shown to affect agonism⁶⁷. To identify statistically coupled sites within this region it will be necessary to exploit a recent structure of the $\alpha 7$ cytoplasmic domain⁶⁸, or perform a traditional statistical coupling analysis based purely on sequence similarity as opposed to the structure-based approach imposed in the present analysis.

Reviewer #2 (Remarks to the Author):

Here Slobodyanyuk et al. present a structure-function study of the pLGIC ELIC. They seek to use structure-based statistical analysis to find a way to mutate ELIC such that ACh switches from being an antagonist to being an agonist. They are ultimately unsuccessful- mutations result in a loss rather than a gain of ACh activity. The conclusion is that regions outside the orthosteric agonist site must be involved in determination of agonist vs. antagonist activity. Overall, the study is clearly written and appears to be carefully performed. The investigation and resolution of why ACh causes potentiation at high doses is satisfying. As currently carried out the study is interesting to me, someone very close to the field, but would be much more interesting if a robust mechanistic finding had been uncovered, rather than just losing ACh binding. As currently presented the reader does not learn much about ELIC or other pLGICs. Negative findings that are well supported are still useful and informative, and I think these are or could be. To increase reader interest, I have the following suggestions for consideration; most are minor text edits.

1. Beginning in Introduction, states are referred to as closed and open. These terms denote conformations, not states; there are multiple closed conformations and perhaps even multiple open conformations. I suggest using resting, activated, and desensitized terminology to describe states, and closed and open to describe conformations.

We agree and appreciate the reviewer's structural perspective. We have now added reference to "activated" and "resting" states. The original choice of words in the first paragraph of our introduction draws heavily upon the wording used in reference 3, which discusses agonism from a classical pharmacology perspective. To preserve connection between this work, and unite the structural and pharmacological perspectives, we have also kept the original wording.

2. End of the introduction. I agree that this is an underappreciated way of stabilizing a non-conducting conformation- essentially through stabilizing a desensitized state without first heavily populating an activated state. There are examples, though, and it might be worth mentioning these. At nAChRs, sazetidine A (PMID: 16857741), AT-1001 (PMID: 25180076, 31488329) are a couple. d-Tubocurarine also likely works this way (very low efficacy agonist = functional antagonist – CHECK Rahman *et al.*). Could mention in discussion if it fits better there and you agree.

While the result is the same, the mechanism we propose for ACh antagonism of ELIC is different from that of putative “silent desensitizers” (and dTc) suggested by the reviewer. In contrast to stabilizing a desensitized state (without appreciable prior activation), we suggest that ACh traps an intermediate conformation along the activation pathway. In other words, ELIC fails to activate in the presence of ACh, but also does not desensitize with ACh bound in the agonist site. ACh occupancy however precludes binding by a more efficacious agonist (in our case cysteamine). The main rationale for this proposed mechanism is based on Pan *et al.*'s X-ray crystal structure of ELIC in complex with ACh, in which it does not appear that ACh traps ELIC in a desensitized state (based on the pore conformation that is more resting-like even though the agonist site is more desensitized/activated-like). As mentioned by the original authors it appears as though ELIC is trapped in a non-conducting intermediate conformation somewhere along the activation pathway (i.e. basically “on the verge of activation”). In any case, our revised manuscript acknowledges this mechanism proposed by the reviewer and takes steps to test this possibility as suggested by the reviewer (see Comment 3 below).

3. If ELIC desensitizes, which the literature suggests it does, ACh maybe acting like the ligands mentioned above, and stabilize a high affinity agonist-bound desensitized-like state. 9' mutations in the pore that alter desensitization have been effective at making it easier for fast desensitizing channels to stay open or, put another way, agonists that cause rapid desensitization (for example in $\alpha 7$) to be more efficacious. A mutant like this might be required to ‘see’ ACh agonist activity.

We agree. The experiment suggested by the reviewer is an excellent one. We installed the well-characterized L9'A substitution into both the wild-type ELIC, as well as the A75D/F133W double mutant background. Despite increasing the potency of cysteamine in both backgrounds, neither L9'A mutant was activated by ACh (see new Figure 6, and new paragraph of the results encompassing lines 387 – 405 of the revised manuscript and pasted below), and thus the L9'A mutation was not enough to “tip the balance” and expose any hidden ACh agonism. While this does not definitively rule out the mechanism proposed by the reviewer, it seems increasingly unlikely in this particular case. At the same time, this finding is consistent with our proposed mechanism where ACh stabilizes a closed intermediate conformation along the activation pathway.

In the end, the goal of the present work is to understand the amino acid differences between ELIC and AChRs that determine their diverging phenotypes with respect to ACh, not simply to convert ACh into an ELIC agonist through a drastic substitution that nature has strongly selected against. Given that a 9' leucine is conserved in both AChRs and ELIC, this residue is not the origin of the diverging phenotypes with respect to ACh in AChRs and ELIC.

Lines 387 – 405 of the revised manuscript:

We investigated whether ACh was trapping ELIC in a desensitized conformation without appreciably, or at least noticeably, activating it. This phenomenon has been observed with various ligands targeting neuronal AChRs^{35–39}, and is thought to underlie the mechanism of *d*-tubocurarine antagonism of the muscle-type AChR^{40–42}. To test this hypothesis, we took

advantage of the well-characterized L240A substitution (L9'A), which maps to the highly conserved M2 transmembrane region lining the ELIC pore. Substitutions at this 9' position in ELIC and AChRs have been shown to dramatically slow agonist-induced desensitization and increase apparent agonist potency⁴³⁻⁴⁷. We evaluated whether the L9'A substitution would similarly slow desensitization and increase potency of ACh for ELIC, thereby exposing any hidden agonism. Consistent with previous data, the desensitization of the L9'A mutant was dramatically slowed in comparison to wild-type (Fig. 6A)^{43,48}, and at the same time the potency of cysteamine was increased (Fig. 6B, Fig. S9, Table 1). We also installed the L9'A substitution into the A75D/F133W double mutant (L9'A + A75D/F133W), which also led to an increase (albeit more modest) in the potency of cysteamine (Fig. 6B, Table 1). We then tested whether ACh could elicit a response in either of these L9'A mutants. Neither mutant produced observable current when increasing concentrations of ACh were applied, but application of 5 mM cysteamine produced a robust peak current in both cases (Fig. 6C,D). In addition, both L9'A mutants showed similar sensitivity to ACh inhibition as their parent channels (Fig. S10). Evidently, the presence of this L9'A substitution is insufficient to expose ACh agonism in either the wild-type ELIC or the A75D/F133W double mutant, suggesting that rapid desensitization upon ACh binding is not the origin for the apparent lack of ACh agonism.

4. Paragraph starting line 226. Please consider toning down the enthusiasm for AChBPs as being useful in probing tertiary/quaternary conformational changes associated with state transitions. To my knowledge, the only parts that move upon agonist binding, based on the available structures, are loop C and to some degree loop F- regions that directly participate in orthosteric ligand binding. Perhaps when coupled to a channel, with mutations in the linkage regions, AChBP can promote gating, but it is not very good at it. To be clear, those soluble receptor homologs are spectacularly useful, but they are not good tools for studying allostery.

We agree. In our revised manuscript we tone down our enthusiasm for AChPBs as a model for probing pLGIC allostery and agonism, and even expose their potential limitations to provide rationale for our new SCA based upon a recent $\alpha 7$ AChR structure.

5. The final section of the discussion ignores desensitization- if one wants to talk about functional efficacy, a two state (resting vs. activated) model is too simple for pLGICs. I could be completely wrong in my idea that perhaps ACh is stabilizing a desensitized state in a real membrane, but the possibility seems worth considering. The bacterial pLGICs are notoriously difficult to get to adopt different conformations for structural biology purposes, especially ELIC- so the channel conformation may not be perfectly informative.

We agree. We now specifically highlight this mechanism of antagonism and acknowledge that we cannot rule it out as a possible mechanism for ACh antagonism of ELIC, especially given the documented difficulties with stabilizing an alternate pore conformation of ELIC.

Lines 498 – 502 of our revised manuscript:

Finally, given the difficulties with stabilizing biologically relevant conformations of ELIC for crystallization⁴³, we cannot exclude the possibility that ACh stabilizes a desensitized state that is not faithfully represented in the ACh-bound ELIC crystal structure. Nevertheless, considering our data with the L9'A mutants, and working under the assumption that the ACh-bound ELIC crystal structure is biologically relevant, we propose that...

Lines 522 – 525 of our revised manuscript:

Although the result is the same, this mechanism of competitive antagonism contrasts with that of classic AChR antagonists, such as *d*-tubocurarine and α -bungarotoxin, that either trap the channel in a desensitized state⁴¹, or arrest it in an antagonized/closed conformation resembling the resting state^{30,49,74}.

Reviewer #3 (Remarks to the Author):

The title of the manuscript “Origins of acetylcholine antagonism in a bacterial pentameric ligand-gated ion channel” in general seems to be correct and justified. To my mind, I would use the word “origin” and would add the restricting abbreviation “ELIC” because acetylcholine is an antagonist of this particular representative of bacterial Cys-loop receptors.

Although the title claims an interest in antagonism, in fact the main task of authors was to modify the ELIC in such a way that acetylcholine would start functioning as an agonist. In spite of the large experimental work this task was not solved and in this way the authors are correct not placing the word “agonist” to the title. The main result that the authors demonstrated is that ELIC mutants A75D and F133W needed considerably higher concentrations to inhibit binding of agonists (thus indicated the points of attachment of the acetylcholine to its antagonist-binding site, and in a way localizing such a sight). However, a double mutant A75D/F133 did not behave as an antagonist at all, not precluding binding of cysteamine, an ELIC agonist – and no explanation of this fact could be provided.

1. AChBP is an excellent model for nicotinic receptors and all other Cys-loop receptors and the authors started with the X-ray structure of the AChBP complex with acetylcholine to choose those mutations which would ensure the ACh agonistic activity on ELIC. As mentioned, it did not work. We should not forget that the ACh binding to AChBP does not produce a current (no channel is available), the binding sites of agonists and antagonists are considerably overlapping, and the main difference in those X-ray structures between complexes with agonists and antagonists is the movement of loop C to the center for the former and for the latter to the periphery. May be the authors could have got better hints for mutations in the ELIC not from the structure of AChBP, but from the X-ray and cryo-EM structures of the LBD complexes with agonists (not compulsory with acetylcholine, for example α 7-AChBP chimera in complex with epibatidine, α 4 β 2 nAChR with bound epibatidine, recent structure of the α 7 receptor in the activated state).

We agree, and now include an additional SCA based upon the recent $\alpha 7$ structure in the activated state mentioned by the reviewer.

2. I was surprised that the authors did not mention that acetylcholine is also an antagonist of the $\alpha 9\alpha 10$ nicotinic receptors and some, not very successful, attempts were made to detect the residues in the binding sites of the $\alpha 9$ subunit responsible for this (although information on the residues involved in binding of α -neurotoxins and various α -conotoxins both on the AChBP and various nAChR subtypes is available).

We have been unable to corroborate the reviewer's assertion that ACh is an antagonist of $\alpha 9\alpha 10$ AChRs. All references we have found suggest that ACh is instead an agonist of these receptors. For this reason, we are unable to address this comment.

3. In the manuscript the authors demonstrated the important role of residues which in the X-ray structure of the AChBP complex with ACh appear quite distant from the bound ACh. Similar conclusions have been also done earlier -see, for example, Inter-residue coupling contributes to high-affinity subtype-selective binding of α -bungarotoxin to nicotinic receptors. Sine SM, Huang S, Li SX, daCosta CJ, Chen L. *Biochem J.* 2013 Sep 1;454(2):311-21.

We agree with this comment, which is similar to that of Reviewer #1's, Comment #2 above. We have now elaborated on this point and included several new references substantiating it.

4. I would appreciate if the authors formulate what is the main conclusion of this manuscript and what is its novelty for the whole family of the Cys-loop receptors

We agree. As requested by the editor and mentioned at the beginning of our Response, we now highlight the significance of our results in several locations and sections throughout the manuscript.

Lines 449 – 455 of our revised manuscript:

Thus, residues deemed important for ACh binding in one background unexpectedly abolish it in another. These results not only demonstrate that additional substitutions are required to convert ACh into an ELIC agonist, but perhaps more importantly that the background upon which residues exist determines their specific contribution to function. This suggests that agonist specificity in pentameric ligand-gated ion channels is highly contingent upon evolutionary history, similar to what has been observed with glucocorticoid receptors^{52,53}.

And,

Lines 519 – 525 of our revised manuscript:

These phenomena reflect a relatively underappreciated mechanism of competitive antagonism, in which *agonists* with appreciable affinity, but negligible efficacy, present as competitive antagonists. Although the result is the same, this mechanism of competitive antagonism contrasts with that of classic AChR antagonists, such as *d*-tubocurarine and α -bungarotoxin, that either trap the channel in a desensitized state⁴¹, or arrest it in an antagonized/closed conformation resembling the resting state^{30,49,74}.

5. Minor comment: No need to introduce to the Abstract “conformational changes indicative of activation”- it was a sort of a hypothesis formulated by other authors.

Thank you. We have left this in the revised manuscript, since this interpretation is important for the mechanism of antagonism we ultimately propose. We have now however mentioned alternate mechanisms in response Reviewer #2's, Comments #2 & #3 above (see new Figure 6, and lines 387 – 405 of our revised manuscript).

END

REVIEWERS' COMMENTS:

Reviewer #2 (Remarks to the Author):

I appreciate the careful revision and feel the manuscript is now suitable for publication.

Reviewer #3 (Remarks to the Author):

I was pleased to read the answers of the authors to the suggestions and critical comments of the three reviewers, including mine. In particular I am glad that they expanded the discussion of possible role of residues distant from the agonist binding site and included the appropriate references. I should also apologize for mistake in one of my comments: I wanted to stress the difference of alpha9/alpha10 receptors from other nAChR subtypes by mentioning that both nicotine and epibatidine are in their case not the agonists but antagonists, but overlooked that instead the word "acetylcholine" was typed. Naturally the authors "were unable to address this comment".